**Data Availability Statement:** All relevant data are within the paper and its Supporting Information file.

# Diagnostic accuracy of a SARS-CoV-2 rapid antigen test among military and civilian personnel of an Air Force airport in central Italy

Paola Verde[1], Cinzia Marcantonio[2◉], Angela Costantino[2◉], Antonio Martina[3◉], Matteo Simeoni[3◉], Stefania Taffon[2], Elena Tritarelli[2], Carmelo Campanella[4], Raffaele Cresta[4], Roberto Bruni[2], Anna Rita Ciccaglione[2], Giulio Pisani[3], Roberto Nisini[2], Enea Spada[2]*

1 Aerospace Medicine Department, Aerospace Test Division, Militay Airport Mario De Bernardi, Pratica di Mare, Rome, Italy, 2 Department of Infectious Diseases, Istituto Superiore di Sanità, Rome, Italy, 3 Center for Immunobiologicals Research and Evaluation, Istituto Superiore di Sanità, Rome, Italy, 4 Institute of Aerospace Medicine, Clinical Analysis and Molecular Biology Laboratory Rome, Rome, Italy

◉ These authors contributed equally to this work.
* enea.spada@iss.it

## Abstract

### Background

Most SARS-CoV-2 rapid antigen detection tests (RADTs) validation studies have been performed on specimens from COVID-19 patients and negative controls or from mostly symptomatic individuals. Herein we evaluated the diagnostic accuracy of AFIAS COVID-19 Ag, hereinafter denominated as AFIAS, during a COVID-19 screening program surveillance testing conducted among personnel of an Italian military airport.

### Methods

Nasopharyngeal swabs (NPSs) were collected from study participants and were analysed by both AFIAS and RT-PCR assay. A questionnaire collecting demographic and exposure data were administered to all participants. AFIAS accuracy parameters including Cohen's kappa (K) were determined.

### Results

Overall, from November 2020 to April 2021, 1294 (NPSs) were collected from 1183 participants (88.6% males, 11.4% females; mean age were 41.3, median age 42). Forty-nine NPSs (3.78%) were positive by RT-PCR, while 54 NPSs were positive by AFIAS. Overall baseline sensitivity, specificity, positive and negative predictive values were 0.633, 0.981, 0.574, 0.985, respectively and K was 0.585 (moderate). AFIAS sensitivity tended to be higher for NPSs with higher viral load. A higher sensitivity (0.944) compared to the overall baseline sensitivity (0.633) was also found for NPSs from participants with COVID-19 compatible symptoms, for which K was 0.891 (almost perfect). Instead, AFIAS sensitivity was quite poor for NPSs from asymptomatic participants. Most false negative NPSs in this group had moderate viral load.

**Funding:** The study was supported in part by the institutional funds of the Istituto Superiore di Sanità, (Fascicolo BA17). The funders had no role in study design, data collection and analysis, decision to publish, or preparation of the manuscript.

**Competing interests:** The authors have declared that no competing interests exist.

## Conclusion

Overall, AFIAS showed high specificity but only moderate sensitivity, mainly because of the high proportion of asymptomatic participants. However, AFIAS showed good sensitivity for NPSs with high viral load and nearly optimal accuracy parameters for NPSs from participants with COVID-19 compatible symptoms. Thus, taking into consideration its performance features, this test can be useful for COVID-19 case identification and management as well as for infection control.

## Introduction

The availability of diagnostic assays for detecting severe acute respiratory syndrome coronavirus 2 (SARS-CoV-2) enabling the identification and effective isolation of COVID-19 cases and the systematic tracing of their close contacts has been critical to mitigate the spread of SARS-CoV-2 infection during the still ongoing pandemic [1, 2].

Nucleic acid amplification techniques (NAATs), like Reverse Transcriptase Real Time Polymerase Chain Reaction (RT-PCR) on nasopharyngeal swabs (NPSs) currently represent the gold standard for SARS-CoV-2 infection diagnosis [3–5]. Unfortunately, performing RT-PCR takes about 3–4 hours and requires special equipment, instruments and skilled laboratory personnel [6]. Furthermore, RT-PCR is costly and subject to reagent and material shortages [7].

Numerous Rapid Antigen Detection Tests (RADTs) have been also developed to diagnose SARS-CoV-2 infection. RADTS have proved particularly suitable as point of care tests (POCT), since they are economical, not requiring the use of particularly complex instrumentation, and have shorter turnaround time (less than 30 minutes). RADTs are primarily lateral flow immunochromatographic assays that detect viral antigen in NPSs by means of a device coated with anti-SARS-CoV-2 antibody [6–17].

Nevertheless, RADTs have demonstrated lower sensitivity for the detection of SARS-CoV-2 in NPSs compared to NAATs. According to a recent systematic review, the average sensitivity and specificity of RADTs were 0.562 and 0.995, respectively [13]. In general, RADTs perform well in NPSs with high viral load or with low cycle threshold (Ct) in RT-PCR for SARS-CoV-2 tests [6–17]. The Ct refers to the number of cycles needed to amplify a fragment of viral genome to reach a threshold level. Ct value provides an indicative measure of viral quantity in the specimen and it is inversely correlated to the original concentration of viral genome in sample tested by RT-PCR. According to the expectations of the European Center of Disease Prevention and Control (ECDC) the implementation of RADTs can serve for the prompt clinical management of cases with COVID-19-compatible symptoms at admission, early detection and isolation of cases and contact tracing [18–20]. In agreement with ECDC and World Health Organization (WHO) [21], Italy has started considering RADTs for diagnosis of symptomatic cases up to five days from symptom onset and, in case of negative result, confirmation with either RADT or molecular tests [22]. According to ECDC, RADTs are also used for case definition, in addition to NAATs [19–21].

The rapid AFIAS COVID-19 Ag (Boditech Med Inc., Chuncheon-si, Gang-won-do, Republic of Korea), hereinafter denominated as AFIAS, is a fluorescence immunoassay designed to detect SARS-CoV-2 antigen in human NPSs within 20 minutes. Thus, although the test requires a fluorescence reader, the running time from sample to result is comparable to traditional RADTs; however, it provides the added value that the result is not subject to the

operator's interpretation. In fact, the positive/negative output of the fluorescence reader is automatic, based on an algorithm comparing the fluorescence obtained from the sample to a cut-off value; in the traditional RADTs, the operator can see more or less well a faint colored band of positivity. This assay has been already included in the Health Security Committee (HSC) Technical Working Group (TWG) common list of antigenic tests, whose results are mutually recognized by European States for public health measures, including issuing European Union (EU) Digital COVID certificate [18, 23].

The aim of this study is to assess the diagnostic accuracy of the AFIAS in comparison to the RT-PCR for detection of SARS-CoV-2 infection in NPSs obtained from military and civilian personnel of a military airport in the metropolitan area of Rome. Differently from similar studies evaluating the performance of RADTs, the study population was unselected and mostly asymptomatic.

## Methods

### Study population

From November 2020 to April 2021, the military and civilian personnel of the "Mario De Bernardi" military airport, located in the metropolitan area of Rome, underwent a screening program to control transmission of SARS-CoV-2 infection in the workplace. The study was part of the public health response to control as soon as possible any outbreak occurring in the military airport (as reported in the Scientific Collaboration Protocol signed by the Experimental Flight Center, Italian Air Force Logistic Command and Istituto Superiore di Sanità on 30 November 2020), with simultaneous evaluation of the RADT used for the screening. According to the national legislation [24–26], testing was performed locally by a first-line rapid antigen test, then positive and negative NPSs were sent to the Istituto Superiore di Sanità (ISS) for confirmation of positive results by NAAT and evaluation of antigen test performance. To assess the accuracy of AFIAS, NPSs were collected from study participants were analysed by both rapid antigen test and RT-PCR assay. All participants were also asked to fill a questionnaire including demographic data, symptoms (if any) and potential exposure to infection (previous COVID-19, contact with proven COVID-19 cases or contact with persons who tested positive to SARS-CoV-2 by molecular or antigenic tests. These data were collected with the aim to characterize the study population and to assess AFIAS accuracy according to presence of symptoms suggestive of COVID-19 and infection exposure. The study protocol was approved and signed by the Scientific Collaboration Partners (i.e. Experimental Flight Center, Italian Air Force Logistic Command and Istituto Superiore di Sanità) on 30 November 2020. Personal data were collected and processed in compliance with EU and Italia legislation [27–29]. Written informed consent was obtained from all participants or the legally authorized representative. Testing, data collection and evaluation of antigen test performance were carried out as public health activities to improve tracing of infected individuals and contacts, with the aim of reducing viral transmission among personnel attending the military airport base.

### Antigen assay for SARS-CoV-2

AFIAS (Boditech Med., Chuncheon-si, Gang-won-do, Republic of Korea) is an immunochromatographic, fluorescence-based rapid antigen test designed to detect the nucleocapsidprotein of SARS-CoV-2 in NP swab specimens. The assay includes an anti-SARS-CoV-2 monoclonal antibody that binds to the viral nucleocapsid protein.

The test was performed according to the manufacturers' instructions. The antigen-antibody interaction leads to a fluorescence signal. Results were interpreted according to the cut-off index (COI), in particular COI <1.0 was interpreted as negative and COI ≥1.0 as positive.

### SARS-CoV-2 molecular detection

The COVID-19 laboratory diagnosis was based on a RT-PCR test (RealStar® SARS-CoV-2 RT-PCR Kit 1.0, Altona Diagnostics) performed on RNA extracts to detect viral RNA. The kit contains all components to enable reverse transcription, PCR-mediated amplification and simultaneous detection of the B-βCoV specific RNA (target E gene) and the SARS-CoV-2 specific RNA (target S gene) as well as the internal control in a single reaction

RNA was extracted from 200μl of NPSs collected in Virus Transport Medium (VTM, Noble Bio, Hwaseong-si, Gyeonggi-do, Republic of Korea) using the QIAamp® MinElute® Virus Spin Kit (QIAGEN, Hilden, Germany).

Every sample was spiked with five μL of RNA Internal Extraction Control (RealStar® SARS-CoV-2 RT-PCR Kit 1.0, Altona Diagnostics) added to the AVL lysis buffer at the first extraction step as a control of nucleic acid extraction step and possible inhibition of RT-PCR. The second part of the nucleic acid purification was performed on the QIAcube instrument (QIAGEN Biotechnology & Life Science).

The extracted RNA was amplified by RT-PCR technology on the Rotor-Gene® instrument (QIAGEN, Hilden, Germany). NPSs with a detectable signal for at least one gene were considered SARS-CoV-2 positive. NPSs with no detectable signal for both genes were considered negative. NPSs with no detectable signal for both target genes and internal control were considered as inconclusive. When we assessed AFIAS accuracy for NPSs from the overall population as well as from the different subgroups of participants, we took also into account the overall median Ct value of both target genes of RT-PCR positive samples. This procedure aimed at assessing AFIAS performance parameters according to different Ct levels (i.e. above or below that median Ct value), has also been suggested by others authors [6, 30–33]. Furthermore, in our accuracy assessment we also considered the definitions for viral load present in the document from the TWG of the EU HSC [18]: very high (Ct≤25), high (Ct 26–30), moderate (Ct >30–36) and low viral load (Ct >36).

### Statistical analysis

Continuous variables were expressed as median and range or as mean ± standard deviation (SD) and differences were compared using the Mann Whitney U test or t-test. Categorical variables were expressed as numbers and percentages and were compared using $X^2$ or Fisher's exact test. A p value < 0.05 was considered significant. Agreement beyond the chance between AFIAS and RT-PCR results was evaluated calculating Cohn's kappa index (K) interpreted according Landis & Koch [34]: <0, no agreement; 0–0.21, slight; 0.21–0.40, fair; 0.41–0.60, moderate; 0.61–0.80, substantial; 0.81–1.0, almost perfect. AFIAS sensitivity, specificity, positive predictive value (PPV), negative predictive value (NPV), positive likelihood ratio (LR+) and negative likelihood ratio (LR-) for NPSs from all study population and for those from the different population subgroups were calculated on contingency tables containing the numbers of each outcome. The confidence intervals (CI) were calculated using the Wilson-Brown method. All analyses were carried out using STATA version 15.0 (College Station, TX, USA)

## Results

### Overall diagnostic accuracy of AFIAS COVID-19 Ag

A total of 1294 NPSs were collected from 1183 individuals. Of them, 88.6% males, 11.4% females. Median and mean age were 42 (range 5–92) and 41.3 (± 10.8), respectively. However, 99.1% of the NPSs were from working age people (18–66 years). One hundred and 11 of the 1183 participants were tested twice and thrice in different occasions, respectively.

**Table 1. Diagnostic accuracy of AFIAS COVID-19 Ag test assessed against RT-PCR assay.**

|  | RT-PCR + | RT-PCR - | Total | Prevalence % | 3.78 (2.88–4.97) |
|---|---|---|---|---|---|
|  |  |  |  | SS | 0.633 (0.493–0.753) |
| AFIAS + | 31 | 23 | 54 | SP | 0.981 (0.972–0.987) |
| AFIAS - | 18 | 1222 | 1240 | PPV | 0.574 (0.441–0.696) |
| Total | 49 | 1245 | 1294 | NPV | 0.985 (0.972–0.990) |
|  |  |  |  | LR+ | 34 (22–54) |
|  |  |  |  | LR- | 0.37 (0.26–0.54) |
|  |  |  |  | % Agreement | 0.969 |
|  |  |  |  | Cohen's kappa | 0.585 (0.470–0.701) |
|  |  |  |  |  | (Moderate) |

AFIAS+, AFIAS COVID-19 Ag positive; AFIAS-, AFIAS COVID-19 Ag negative; COI, Cut Off Index, LR+, positive likelihood ratio; LR-, negative likelihood ratio; NPV, negative predictive value; PPV, positive predictive value; RT-PCR+, Real-time RT-PCR positive; RT-PCR-, Real-time RT-PCR negative; SS, sensitivity; SP, specificity.

Of the 1294 NPSs assayed, 49 (75.5% males; mean age, 42.85 ± 13.8) were positive by RT-PCR assay indicating a SARS-CoV-2 prevalence of 3.78%. Fifty-four NPSs (4.17%) were positive by AFIAS, while 31 NPSs (2.39%) resulted positive by both these tests (true-positive) (Table 1). Eighteen and 23 NPSs resulted false-negative (RT-PCR+/AFIAS-) and false-positive (RT-PCR-/AFIAS+), respectively. The number of true-negative NPSs (RT-PCR-/AFIAS-) was 1222. Overall, AFIAS showed a good SP and NPV but a moderate SS and PPV. Percent agreement between the results of the AFIAS results and those of the reference test was nearly 0.97, but K was 0.585 (Table 1).

Analysing in depth the comparison between the two tests (Table 2) – taking also into account the overall median Ct value of both target genes of RT-PCR positive NPSs (i.e., 22.34)–it was evident that positive NPSs with Ct ≤22.34 were detected by AFIAS with higher sensitivity compared to NPSs with Ct >22.34 (Ct ≤22.34, SS = 22/25 = 0.880 *vs.* Ct > 22.34, SS = 9/24 = 0.375; p = 0.00001).

In agreement with this latter finding, Fig 1 shows as the Ct values for each of the two PCR target genes were significantly higher among NPSs with false-negative results than among NPSs with true-positive Ag results (S gene Ct/AFIAS Ag- *vs* S gene Ct/AFIAS Ag+, p = 0.00026; E gene Ct/AFIAS Ag- *vs* E gene Ct/AFIAS Ag+, p<0.00001).

Table 3 shows the distribution of the 49 RT-PCR positive NPSs, according to HSC TWG viral load cut-off and AFIAS results (i.e. true-positive or false-negative).

**Table 2. AFIAS COVID-19 Ag test sensitivity according to the median RT-PCR Ct value (22.34).**

|  | RT-PCR + | ≤22.34 RT-PCR + | >22.34 RT-PCR + | RT-PCR - | Total |
|---|---|---|---|---|---|
| AFIAS + | 31 | 22 | 9 | 23 | 54 |
| AFIAS - | 18 | 3 | 15 | 1222 | 1240 |
| Total | 49 | 25 | 24 | 1245 | 1294 |

RT-PCR+ 49 NPSs (97 S and E gene positive amplifications); median Ct, 22.34; range Ct, 12.12–37.76

Ct ≤22.34: 22 NPSs RT-PCR+/AFIAS+; 3 NPSs RT-PCR+/AFIAS- SS = 22/25 = 0.880

Ct >22.34: 9 NPSs RT-PCR+/AFIAS+; 15 NPSs RT-PCR+/AFIAS- SS = 9/24 = 0.375

AFIAS +, AFIAS COVID-19 Ag positive; AFIAS -, AFIAS COVID-19 Ag negative; COI, Cut Off Index, LR+, positive likelihood ratio; LR-, negative likelihood ratio; NPV, negative predictive value; NPSs, Nasopharyngeal swabs; PPV, positive predictive value; RT-PCR+, Real-time RT-PCR positive; RT-PCR-, Real-time RT-PCR negative; SS, sensitivity; SP, specificity.

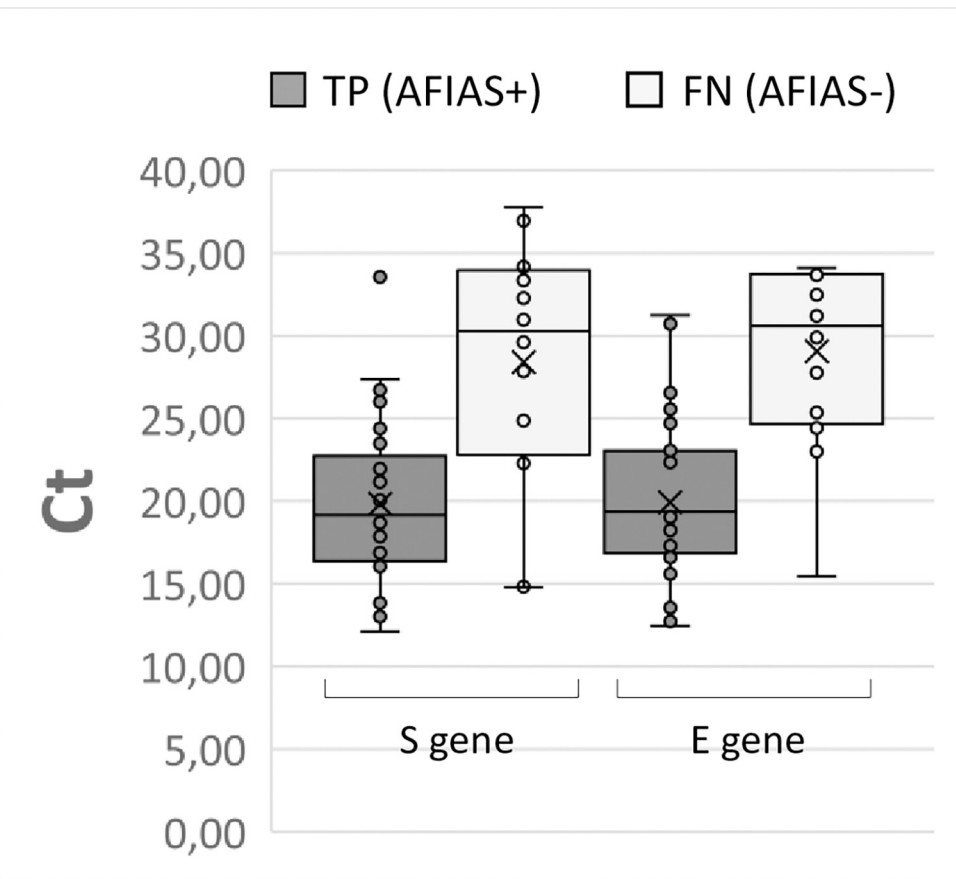

**Fig 1. Box-plot distribution of Ct values obtained by RT-PCR targeted to the viral S gene and E gene from NP swab NPSs, broken down according to the result obtained by AFIAS COVID-19 Ag rapid test.** AFIAS+: true-positive, TP; AFIAS-: false-negative, FN.

## Diagnostic accuracy of AFIAS COVID-19 Ag according to presence or absence of symptoms suggestive of COVID-19

Apart from 73 NPSs taken from participants with symptoms suggestive of COVID-19 at the time of testing, the remaining 1221 NPSs were from asymptomatic participants

The diagnostic accuracy parameters for NPSs from symptomatic and asymptomatic participants are shown in Table 4.

In this table, RT-PCR positive NPSs from both asymptomatic and symptomatic participants were further stratified in two sub-groups according to the overall median Ct value of both target genes.

**Table 3. RT-PCR positve NPSs according to HSC TWG Ct and AFIAS result.**

| Ct | Viral load | NPSs Ct range | N° RT-PCR + | N° TP | N° FN |
|---|---|---|---|---|---|
| ≤25 | Very high | 12.12–24.86 | 32 | 26 | 6 |
| 26–30 | High | 25.07–29.90 | 5 | 3 | 2 |
| >30–36 | Moderate | 30.02–34.10 | 11 | 2 | 9 |
| >36 | Low | 37.62 | 1 | 0 | 1 |

AFIAS, AFIAS COVID-19 Ag assay, NPSs, Nasopharingeal samples; N°, numero of samples

RT-PCR+, Real-time RT-PCR positive; FN, false-negative; TP, true-positive.

**Table 4. Diagnostic accuracy of AFIAS COVID-19 Ag test according to presence or absence of COVID-19 symptoms.**

| | NPSs from participants with symptoms suggestive of COVID-19 | | | | | NPSs from participants without symptoms suggestive of COVID-19 | | | | |
|---|---|---|---|---|---|---|---|---|---|---|
| | | <18.08[a] | >18.08 | | | | ≤26.00[b] | >26.00 | | |
| | RT-PCR+ | RT-PCR+ | RT-PCR+ | RT-PCR- | Total | RT-PCR+ | RT-PCR+ | RT-PCR+ | RT-PCR- | Total |
| AFIAS + | 17 | 10 | 7 | 2 | 19 | 14 | 11 | 3 | 21 | 35 |
| AFIAS - | 1 | 0 | 1 | 53 | 54 | 17 | 5 | 12 | 1169 | 1186 |
| Total | 18 | 10 | 8 | 55 | 73 | 31 | 16 | 15 | 1190 | 1221 |
| | | 95% CI | | | | | 95% CI | | | |
| Prevalence | 24.66 | 16.20–35.64 | | | | 2.54 | 1.79–3.58 | | | |
| SS | 0.944 | 0.724–0.990 | | | | 0.452 | 0.292–0.622 | | | |
| SP | 0.964 | 0.877–0.990 | | | | 0.982 | 0.973–0988 | | | |
| PPV | 0.895 | 0.686–0.971 | | | | 0.400 | 0.255–0.564 | | | |
| NPV | 0.981 | 0.923–0.997 | | | | 0.986 | 0.977–9.991 | | | |
| LR+ | 26 | 6.63–102 | | | | 26 | 14–45 | | | |
| LR- | 0.06 | 0.01–0.39 | | | | 0.56 | 0.41–077 | | | |
| % Agreement | 0.959 | 0.886–0.986 | | | | 0.969 | 0.958–0.977 | | | |
| Cohen's Kappa | 0.891 | 0.771–1.000 | | Almost perfect | | 0.408 | 0.256–0.561 | | Moderate | |

AFIAS +, AFIAS COVID-19 Ag positive; AFIAS -, AFIAS COVID-19 Ag negative; COI, Cut Off Index, LR+, positive likelihood ratio; LR-, negative likelihood ratio; NPSs, nasopharyngeal swabs, NPV, negative predictive value; PPV, positive predictive value; RT-PCR+, Real-time RT-PCR positive; RT-PCR-, Real-time RT-PCR negative; SS, sensitivity; SP, specificity.

[a] median Ct value for NPSs from RT-PCR positive participants with symptoms suggestive of COVID-19.

[b] median Ct value for NPSs from RT-PCR positive participants without symptoms suggestive of COVID-19.

As expected, the prevalence of positive RT-PCR results in NPSs from symptomatic partici- pants was nearly tenfold higher than in NPSs from asymptomatic participants (p<0.0001) and, again as expected, median Ct value for both RT-PCR target genes was significantly lower in NPSs from symptomatic participants than in NPSs from asymptomatic ones (18.08 vs. 26.00; p<0.00001). Optimal accuracy parameter values were found for NPSs from symptom- atic participants, also showing an almost perfect agreement beyond the chance (K = 0.891) (Table 3). On the contrary, among NPSs from asymptomatic participants, despite specificity and NPV were good, sensitivity and PPV were quite low and a barely moderate K (0.408) was found. The low PPV found for NPSs from asymptomatic participants, also considering the high specificity (0.986) shown by the AFIAS, indicated that it was clearly influenced by the low prevalence among them of RT-PCR positive results (nearly tenfold lower than in NPSs from symptomatic participants; p<0.0001).

Fig 2A shows the Ct value distribution for S and E genes in RT-PCR positive NPSs from participants with and without symptoms suggestive of COVID-19 at time of testing. From Fig 2A it is evident that the difference in sensitivity (Table 3: 0.944 vs. 0.452) between these two groups of NPSs was clearly linked to the different distribution of Ct values in the two groups. Median Ct values in NPSs from symptomatic participants were significantly lower than in NPSs from asymptomatic participants for both the S and E genes (S gene: 18.52 vs. 25.43; E gene: 18.09 vs. 26.00).

Fig 2B and Table 3 allow to analyse in more detail the difference in sensitivity and other accuracy parameters of the AFIAS between NPSs from symptomatic and asymptomatic partic- ipants. Among the 18 RT-PCR positive NPSs from symptomatic participants, only one of them (1/18) was false-negative and, not surprisingly, all the 18 RT-PCR positive NPSs had very high viral load, including the only false-negative sample (Fig 2B). On the contrary, among the 31 RT-PCR positive NPSs from asymptomatic participants more than half (17/31) were false-

**A**

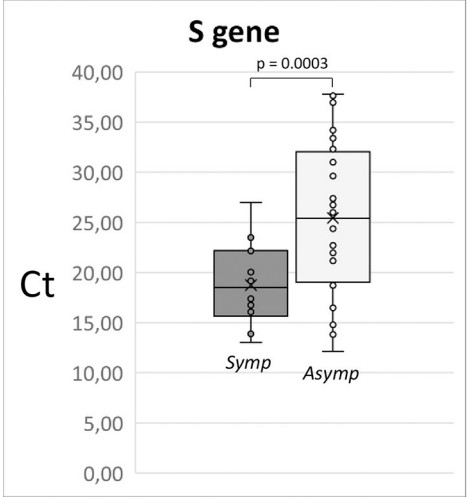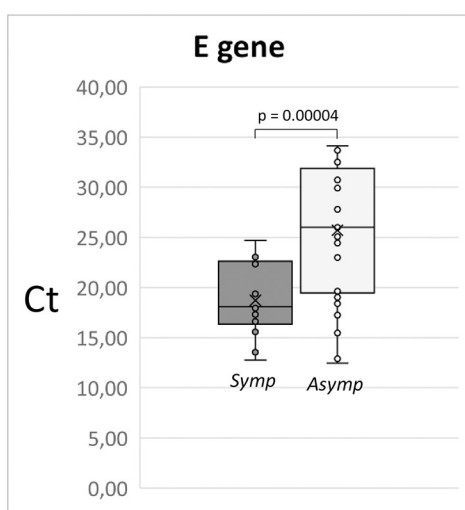

**B**

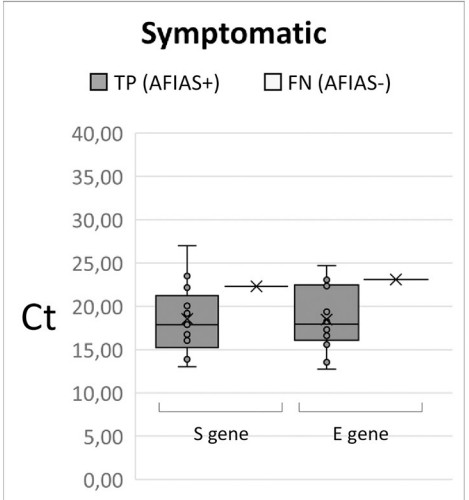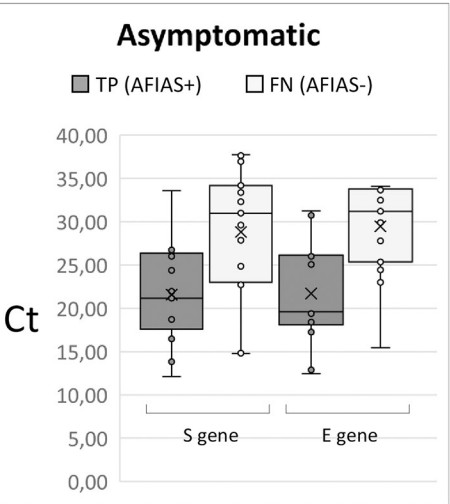

**Fig 2.** (A) Box-plot distribution of Ct values obtained by RT-PCR targeted to the viral S gene (left) and E gene(right) from NP swab NPSs from symptomatic (Symp) and asymptomatic (Asympt) individuals. (B) Box-plot distribution of Ct values from symptomatic and asymptomatic individuals, broken down according to the result obtained by AFIAS COVID-19 Ag rapid test. AFIAS+: true-positive, TP; AFIAS-: false-negative, FN.

negative, and, importantly, nearly 60% of them (10/17) had Ct>30 (i.e. moderate/low viral load).

## Diagnostic accuracy of AFIAS COVID-19 Ag test in specimens from participants more likely to test positive for COVID-19

Among all collected NPSs, 57 were from participants reporting a previous positive SARS-CoV-2 RT-PCR test (A), 94 from those who had had contact with an ascertained COVID-19

case (confirmed by RT-PCR) (B) and 150 from participants who had had contact with people reporting a positive for SARS-CoV-2 test (not specified if molecular or antigenic) (C).

Diagnostic accuracy parameters for the above subgroups (hereafter referred to as A, B, C) are shown in Table 5. The prevalence of RT-PCR positive NPSs was higher among subgroup A respect to the other two groups, but the differences were not statistically significant. However, NPSs from subgroup A showed the lowest sensitivity and NPV, with a relevant proportion of false-negative results (8/13). As expected, false negative results were more frequent in NPSs with a Ct value higher than the median Ct (29.60).

The NPSs included in subgroup A had a significant higher median Ct value compared to specimens from subgroup B and C (p = 0.0029 and p = 0.0048, respectively). Diagnostic accuracy parameters were better for these two latter subgroups as also demonstrated by their K indicating a substantial agreement beyond the chance.

## Discussion

Prompt and accurate testing is key for identification and clinical management of COVID-19 cases and for surveillance, control and prevention of SARS-CoV-2 infection [1, 2].

Since December 2020, the EU case definition for COVID-19 includes the detection of antigens in the clinical specimens and therefore the use of RADTs as a diagnostic method [19, 20]. Currently available RADTs show a variable but lower sensitivity compared to RT-PCR test, while their specificity is generally high [6–17]. It is worthy of note that RADTs are sensitive enough to detect cases with a high viral load [i.e. pre-symptomatic and early symptomatic cases (up to five days from symptom onset); or low RT-PCR Ct value (≤25)]. Such cases likely account for a significant proportion of SARS-CoV-2 transmission events [20].

Most of RADTs validation studies were performed in cohorts of symptomatic participants, while a limited number of these studies were performed in asymptomatic or predominantly asymptomatic individuals [35–39], to support the use of these tests in mass screening and epidemiological surveillance.

According to a more recent systematic review [40], the average SS in 37 studies carried out in symptomatic individuals was 0.720 [95% Confidence Interval (CI) 0.637–0.790] while that in 12 studies in asymptomatic individuals was 0.581 (95% CI, 0.402–0.701).

Overall, in our study, in which SARS-CoV-2 infection prevalence was 3.8%, AFIAS showed a good specificity (0.981) but a moderate sensitivity (0.633). However, we observed a quite good sensitivity (0.880) for RT-PCR positive NPSs with a Ct value lower than the overall median Ct value of both target genes (Ct≤22.34, i.e. corresponding to a "very high viral load" level) (Tables 2 and 3). An increased sensitivity with higher viral load was also found when results were analysed according to the presence of symptoms (0.944, Table 4). Sensitivity and specificity figures reported in a systematic revision for rapid antigen immune-fluorescent assays were higher than those found in our work for AFIAS [13, 41–43]. However, this is not at odds with the overall results of our study because the respiratory samples tested in the studies considered in that systematic revision were from individuals with fever or with respiratory symptoms suggestive of COVID-19 [41], or from suspected COVID-19 cases [42, 43]. The overall moderate AFIAS's sensitivity in our study could undoubtedly be influenced by the larger proportion of NPSs from asymptomatic participants, which accounted for 94.3% of the total NPSs and 63% of the RT-PCR positive NPSs. In fact, NPSs from the asymptomatic participants showed both a higher median Ct value and a wider range of Ct than NPSs from the symptomatic participants, ultimately resulting in a significant greater proportion of false-negative results (17/31, 54.8% *vs.* 1/18, 5.6%; p = 0.0005) (Table 3).

**Table 5. Diagnostic accuracy of AFIAS COVID-19 Ag test in participants with previous positive molecular test or high-risk contact.**

| | (A) NPSs from participants with previous positive SARS-CoV-2 PCR test | | | | | (B) NPSs from participants reporting contact with COVID-19 case[a] | | | | | (C) NPSs from participants reporting contact with people tested positive for SARS-CoV-2[a] | | | | |
|---|---|---|---|---|---|---|---|---|---|---|---|---|---|---|---|
| | RT-PCR+ | <29.60[b] RT-PCR+ | >29.60 RT-PCR+ | RT-PCR- | Total | RT-PCR+ | <22.06[c] RT-PCR+ | >22.06 RT-PCR+ | RT-PCR- | Total | RT-PCR+ | <20.30[d] RT-PCR+ | >20.30 RT-PCR+ | RT-PCR- | Total |
| AFIAS + | 5 | 4 | 1 | 2 | 7 | 10 | 7 | 3 | 2 | 12 | 12 | 10 | 2 | 2 | 14 |
| AFIAS - | 8 | 3 | 5 | 42 | 50 | 3 | 0 | 3 | 79 | 82 | 6 | 1 | 5 | 130 | 136 |
| | 13 | 7 | 6 | 44 | 57 | 13 | 7 | 6 | 81 | 94 | 18 | 11 | 7 | 132 | 150 |
| | | 95% CI | | | | | 95% CI | | | | | 95% CI | | | |
| Prevalence | 22.81 | 13.84–35.21 | | | | 13.82 | 8.26–22.24 | | | | 12.0 | 77.3–18.2 | | | |
| SS | 0.385 | 0.117–0.6645 | | | | 0.769 | 0.497–0.918 | | | | 0.667 | 0.437–0.837 | | | |
| SP | 0.954 | 0.849–0.987 | | | | 0.975 | 0.914–0.993 | | | | 0.985 | 0.946–0.996 | | | |
| PPV | 0.714 | 0359–918 | | | | 0.833 | 0.552–0.953 | | | | 0.857 | 0.600–0.960 | | | |
| NPV | 0.840 | 0.715–0.917 | | | | 0.963 | 0.898–0.987 | | | | 0.956 | 0.907–0.980 | | | |
| LR+ | 8.46 | 11.85–39 | | | | 31 | 7.68–126 | | | | 44 | 11–181 | | | |
| LR- | 0.64 | 0.42–1.00 | | | | 0.24 | 0.09–0.64 | | | | 0.34 | 0.18–0.65 | | | |
| % Agreement | 0.824 | 0.706–0.902 | | Moderate | | 0.946 | 0.881–0.977 | | Substantial | | 0.946 | 0.898–0.973 | | Substantial | |
| Cohen's Kappa | 0.405 | 0.113–0697 | | | | 0.769 | 0.576–0.963 | | | | 0.761 | 0.538–0.903 | | | |

AFIAS +, AFIAS COVID-19 Ag positive; AFIAS -, AFIAS COVID-19 Ag negative; COI, Cut Off Index, LR+, positive likelihood ratio; LR-, negative likelihood ratio; NPSs, nasopharyngeal swabs; NPV, negative predictive value; PPV, positive predictive value; RT-PCR +, Real-time RT-PCR positive; RT-PCR-, Real-time RT-PCR negative; SS, sensitivity; SP, specificity.

[a] contact within the last 14 days

[b] median Ct value for NPSs from RT-PCR positive participants with previous positive SARS-CoV-2 PCR test

[c] median Ct value for NPSs from RT-PCR positive participants reporting contact with COVID-19 case (confirmed by RT-PCR)

[d] median Ct value for NPSs from RT-PCR positive participants reporting contact with people tested positive for COVID-19 by molecular or rapid antigen test.

As regards the other accuracy parameters in NPSs from symptomatic participants, the AFIAS showed very good performance parameters (SP 0.964, PPV 0.895, NPV 0.981) and K was 0.891. As expected, in this subgroup the SARS-CoV-2 prevalence was very high (24.7%), while among NPSs from asymptomatic participants the prevalence was about ten-fold lower, leading to a lower PPV than in symptomatic participants' NPSs (0.400 vs 0.895).

For NPSs from participants with previous positive SARS-CoV-2 PCR test, for which the infection prevalence was similar (22.8%) to that found for NPSs from symptomatic participants (24.7%), AFIAS sensitivity was very low (0.385) with a barely moderate K. These findings indicated that most NPSs from this group were indeed from non-recent infections, according to the observed relatively low viral load (median Ct: 29.60), leading to high proportion of false-negative results (8/13, 61.5%).

The link between viral load and sensitivity was further confirmed when the findings obtained for NPSs from participants with a previous positive SARS-CoV-2 PCR test (prevalence, 22.80%) were compared to those found for NPSs from participants reporting a contact with a COVID-19 case (prevalence, 13.82%) and or a contact with people tested positive for SARS-CoV-2 within the previous 14 days (prevalence, 12.00%) (Table 4, subgroups A, B and C). In fact, in both these latter subgroups of participants, sensitivity was higher (with a substantial K) than in participants with a previous positive SARS-CoV-2 PCR test. Indeed, in both these groups the median viral load (inversely related to the median Ct value) was higher, likely indicating a more recent infection.

Comparison of the PPV from the three above-mentioned groups shows that the expected influence of prevalence on PPV is counterbalanced by the viral load: in fact, despite the prevalence decreases (22.81%, 13.82% and 12.00%), the PPV increases (71.4%, 83.3% and 85.7%), with a minor role of specificity (95.4%, 97.5% and 98.5%) (see Table 4).

Other studies have previously assessed the diagnostic accuracy of AFIAS [10, 11, 14]. A Korean study performed on clinical specimens from 38 adult COVID-19 and 122 non-COVID-19 patients showed an excellent SP (0.987–0.989), but the sensitivity was good only for NPSs with high viral load (0.913–1.00 in NPSs with Ct <25) [10]. In the present study, the sensitivity was 0.813 (26/32) in NPSs with Ct ≤25. This difference might be due to different distribution of Ct values in NPSs with Ct ≤25 between the two studies, but this hypothesis could not be verified as detailed information was not reported in the Korean study. Another study assessed the accuracy of AFIAS COVID-19 Ag in a mass screening carried out among unselected paediatric patients. Overall sensitivity and specificty were 86.4% and 98.3%, respectively [11]. Also in that study, information about distribution of Ct values as well as presence of symptoms were not reported, precluding any detailed comparison with results of the present study. Finally, another Italian study evaluating the performance of three different RADTs, assessed the accuracy of AFIAS by testing 81 known SARS-COV-2 positive and negative respiratory samples. Sensitivity and specificity found in this study were 0.375 a 0.100 respectively [14].

Our study had some limitations. Precise information on the time to onset of symptoms was not available. Similarly, participants were asked if they had had contact with cases or people with previous positive tests within 14 days earlier, but precise information on contact time point was not collected.

In conclusion, AFIAS COVID Ag, as many other RADTs, showed an overall high specificity, a good sensitivity for NPSs with high viral load and nearly optimal accuracy parameters for participants with COVID-19 compatible symptoms. Hence, taking into consideration its performance features, in high prevalence settings/population this test might be used for COVID-19 case identification and management, as well as for infection control.

## Supporting information

**S1 Data.**
(XLSX)

## Author Contributions

**Conceptualization:** Paola Verde, Roberto Bruni, Anna Rita Ciccaglione, Giulio Pisani, Roberto Nisini, Enea Spada.

**Data curation:** Cinzia Marcantonio, Angela Costantino, Elena Tritarelli, Roberto Bruni, Anna Rita Ciccaglione, Enea Spada.

**Formal analysis:** Roberto Bruni, Anna Rita Ciccaglione, Enea Spada.

**Funding acquisition:** Paola Verde, Anna Rita Ciccaglione, Giulio Pisani, Roberto Nisini.

**Investigation:** Cinzia Marcantonio, Angela Costantino, Antonio Martina, Matteo Simeoni, Stefania Taffon, Elena Tritarelli, Carmelo Campanella, Raffaele Cresta.

**Methodology:** Cinzia Marcantonio, Angela Costantino, Antonio Martina, Matteo Simeoni, Stefania Taffon, Elena Tritarelli, Carmelo Campanella, Raffaele Cresta.

**Project administration:** Paola Verde, Anna Rita Ciccaglione, Giulio Pisani, Roberto Nisini.

**Resources:** Paola Verde, Anna Rita Ciccaglione, Giulio Pisani, Roberto Nisini.

**Supervision:** Paola Verde, Roberto Bruni, Anna Rita Ciccaglione, Giulio Pisani, Enea Spada.

**Validation:** Paola Verde, Anna Rita Ciccaglione, Giulio Pisani.

**Visualization:** Roberto Bruni, Enea Spada.

**Writing – original draft:** Enea Spada.

**Writing – review & editing:** Paola Verde, Roberto Bruni, Anna Rita Ciccaglione, Giulio Pisani, Roberto Nisini, Enea Spada.

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
