## [Decision Letter · Decision Letter 0]

22 Jul 2022

PONE-D-22-13832Diagnostic accuracy of a SARS-CoV-2 rapid antigen test among military and civilian personnel of an Air Force airport in central ItalyPLOS ONE

Dear Dr. Spada,

Thank you for submitting your manuscript to PLOS ONE. After careful consideration, we feel that it has merit but does not fully meet PLOS ONE’s publication criteria as it currently stands. Therefore, we invite you to submit a revised version of the manuscript that addresses the points raised during the review process.

The work described in this manuscript is valid even though the study is not original and the results were predictable based on previous literature. However, the manuscript should be revised for clarity, internal congruency and completeness of details and information. The reviewers identified critical issues and concerns that should be addressed point-by-point in the revised manuscript. If you decide to re-submit the manuscript, I'd also invite you to consider reducing the number of tables as suggested by one reviewer.   

We look forward to receiving your revised manuscript.

Kind regards,

Luisa Gregori

Academic Editor

PLOS ONE

Journal Requirements:

Reviewers' comments:

Reviewer's Responses to Questions

**Comments to the Author**

1. Is the manuscript technically sound, and do the data support the conclusions?

Reviewer #1: Partly

Reviewer #2: Yes

Reviewer #3: Yes

2. Has the statistical analysis been performed appropriately and rigorously? 

Reviewer #1: N/A

Reviewer #2: Yes

Reviewer #3: Yes

3. Have the authors made all data underlying the findings in their manuscript fully available?

Reviewer #1: Yes

Reviewer #2: Yes

Reviewer #3: Yes

4. Is the manuscript presented in an intelligible fashion and written in standard English?

Reviewer #1: No

Reviewer #2: Yes

Reviewer #3: Yes

5. Review Comments to the Author

Reviewer #1: The authors tested from November 2020 to April 2021, 1294 nasopharyngeal swab samples from 1183 participants with the rapid antigen test AFIAS COVID-19 Ag, a fluorescence-based rapid antigen test. This is a not wide-spread rapid antigen test, not easy to work with as the development of the final reaction is fluorescence and a little instrumentation is required for reaction reading. Results were compared with standard RT-PCR (gene E and N, by Altona) according to the presence of symptoms and the level of viral load as expressed by the Ct value as a proxy. Testing was performed within the military and civilian personnel of a military airport, located in the metropolitan area of Rome, who underwent a screening program to control transmission of SARS-CoV-2 infection, Prevalence rate of Covid-19 in the studied population: 3.78%.

Forty-nine samples (3.78%) were positive by RT-PCR (32 of them with a high viral load: Ct range 12.12-24.86), while 54 samples were positive by AFIAS COVID-19 Ag; 18 samples were false negative by AFIAS and 23 were false positive. AFIAS overall sensitivity, specificity, positive and negative predictive values were 0.633, 0.981, 0.574, 0.985, respectively with only a moderate level of concordance with RT-PCR. These figures increased when comparison was made in samples with a high viral load (Ct ≤22). Similar results were achived if AFIAS COVID-19 Ag was compared with RT-PCR results in symptomatic patients and this is certainly a results of higher viral loads in these patients. Instead, and as expected by the scientific literature, test sensitivity was poor for samples from asymptomatic participants and for viral load CT >30.

In conclusion, the authors stated that AFIAS COVID Ag showed high specificity but only moderate sensitivity in the screened population where the prevalence of COVID-19 was, by the way, low. However, and again as expected, the assay showed good sensitivity for samples with high viral load and in participants with COVID-19 compatible symptoms. Thus, in high prevalence setting this test can be useful for COVID-19 case identification and management at a point-of-care level.

The work is in line with many other papers on rapid antigen testing in COVID-19 pandemic showing that the sensitivity of these type of tests is higher in symptomatic than in asymptomatic patients and that using the Ct value as a proxy for viral load, the sensitivity increases with the increasing of viral loads. All these data have already been acknowledged by the scientific and medical community, the only reason to ask a re-submission from the authors is that there are not so many data on the type of rapid antigen test used by them, e.g. AFIAS COVID-19 Ag and it would be certainly usefull to add more data to the existing ones. The good correlation with RT-PCR is only for Ct <22, which means that the technique of this rapid test e.g. fluorescence, is not as sensitive as the one of other rapid test working with fluorescence in the same context (Dinnes J, Deeks JJ, Adriano A, Berhane S, Davenport C, Dittrich S, et al. Rapid, point-of-care antigen and molecular-based tests for diagnosis of SARS-CoV-2 infection. Cochrane Database Syst Rev. 2020 Aug 26;8(8). There are no comments by the authors on this important issues.

Therefore, my suggestion is to re-write it as short report, making the study more synthetic, technical and readable, cutting down the number of tables to the essential (just one as the overall evaluation and one more according to the Ct level). The Ct level chosen for statistical analysis should be the same across all the study (e.g. abstract ct<25, table Ct< 22.34, text Ct <22), as there are only 49 positive samples by RT-PCR, 18 of them negative by the antigen test (the authors are working on 31 positive concordant samples, which is not that much). Therefore, there is no clue in making several subsets of positive samples/patients (and each one corresponds to a table!) since they are just few positive samples by the gold standard RT-PCR. This gives the work a strong reading difficulty and hard to follow the analysis. Moreover, it is not clear if the statistical analysis is run on sample or patients (1294 vs 1183, text vs. tables).

Reviewer #2: The authors evaluated a SARS-CoV-2 rapid antigen detection test (RADT) while using RT-PCR as the gold standard. The works were scientifically sound. However, mistakes and confusions were found and I have several queries for the current version of the manuscript. I hope the authors will find them useful to revise it so that the quality could be improved.

- the research gap was not well defined:

(1) lines 95-100: the authors have to explain the reasons for choosing ‘military and civilian personnel of a military airport in the metropolitan area of Rome’ as the subjects

(2) lines 69, 87-88, 114-121: the authors stated that RADTs are easy to use since equipment was not required, however, the authors employed the fluorescence based RADT in the study. This test requires an equipment to read the test results. It seems that the introduction was discordant to the method used.

- line 172: you have to mention the meaning of the cut-off value Ct 22.34, readers will not know that unless they went through the table 1b

- lines 220-221, 272-273, table 3, footnote a: you have to define group B and group C clearly. I still do not know the differences between groups B and C after going through these explanations. I only know that participants in groups B and C had contact history with COVID-19 patients within the last 14 days. Participants in groups B will be more likely to be detected by RADT than those in group C. If I understand correctly, groups B participants had contact history with COVID-19 patients tested positive by RADT while groups C participants had contact history with COVID-19 patients tested positive by RT-PCR. Please confirm my speculation.

- the authors analyzed the RADT results according to the four different parameters: (1) EU HSC (2) RT-PCR results (3) presence of symptoms (4) contact history with COVID-19 cases. It means that many different cut-offs were used:

(1) EU HSC cut-offs: <25, 26-30, 30-36, >36 (lines 141-142)

(2) median RT-PCR: 22.34

(3) symptomatic patients: 18.08

(4) asymptomatic patients: 26.00

For (1), a table is preferred rather than just describing the results in text only (lines 179-182).

 

Minor comments:

- avoid creating unnecessary abbreviations if the fluency is not improved, it is not inconvenient to spell the terms SS (line 73) and SP (line 75) in full as sensitivity and specificity respectively. I cannot see those terms will either save the word counts or make the manuscript more neat and tidy. In addition, these two terms are not commonly used by other research groups. Readers have to memorize them throughout the manuscript. It is easy to create confusion. On the other hand, the term ‘N’ referring nucleocapsid protein created (line 117) has not been used in subsequent texts. All these kinds of arrangements make this manuscript quite unprofessional.

- there were only two different assays, RT-PCR and RADT, it is not necessary to create another term, index test in lines 34, 41, 44, 49, 169, 200, 211, 254, 260. As you define RADT at the beginning, you can either use this term throughout the manuscript or make a short form for the RADT that your performed in your study, ‘AFIAS’ to refer ‘AFIAS COVID-19 Ag’.

Reviewer #3: In this manuscript, Authors report the results of a screening program to control transmission of SARS-CoV-2 infection in the workplace. The study was conducted from November 2020 to April 2021 on the personnel of a military airport in Rome. Tests were performed with immunochromatographic fluorescence-based rapid antigen test designed to detect the nucleocapsid protein (N) of SARS-CoV-2 in nasopharyngeal swab specimens.

The study was conducted appropriately. Nevertheless, an important limitation is that the study was carried out almost a year and a half ago: the epidemiological situation and the variants circulating today are different. Moreover, the topic of the article appears to be of limited interest since it has been extensively covered in similar published works regarding the same rapid antigen test and others similar. In addition to the two studies already mentioned in the manuscript’s discussion, some other examples are reported below:

- Baccani I, Morecchiato F, Chilleri C, Cervini C, Gori E, Matarrese D, Bassetti A, Bonizzoli M, Mencarini J, Antonelli A, Rossolini GM. Evaluation of Three Immunoassays for the Rapid Detection of SARS-CoV-2 antigens. Diagn Microbiol Infect Dis. 2021 Oct;101(2):115434. doi: 10.1016/j.diagmicrobio.2021.115434. Epub 2021 May 21. PMID: 34174523; PMCID: PMC8137375.

- Parvu, V.; Gary, D.S.; Mann, J.; Lin, Y.C.; Mills, D.; Cooper, L.; Andrews, J.C.; Manabe, Y.C.; Pekosz, A.; Cooper, C.K. Factors that influence the reported sensitivity of rapid antigen testing for SARS-CoV-2. Front. Microbiol. 2021, 12, 714242.

- Filchakova, O.; Dossym, D.; Ilyas, A.; Kuanysheva, T.; Abdizhamil, A.; Bukasov, R. Review of COVID-19 testing and diagnostic methods. Talanta 2022, 244, 123409.

- Bruzzone, B.; De Pace, V.; Caligiuri, P.; Ricucci, V.; Guarona, G.; Pennati, B.M.; Boccotti, S.; Orsi, A.; Domnich, A.; Da Rin,G.; et al. Comparative diagnostic performance of rapid antigen detection tests for COVID-19 in a hospital setting. Int. J. Infect.Dis. 2021, 107, 215-218.

Here below, Authors can find a list of revisions that need to be addressed in order to improve the quality of the manuscript.

Abstract

- line 38: median age is missing from results

- line 41-44: it is not so clear what concept the authors want to express

Methods

- the method of enlistment is not specified (they were volunteers?)

- It is not specified how was determined the number of people to be enlisted (only time criteria?)

- line 104-106: we suggested to specify the use that will be made of the data contained in the questionnaires

- viral variants circulating at the time of the study are never mentioned in the study. Therefore, data on their detection capabilities for the test under consideration are missing.

- line 139-140: it would be helpful to specify the number of samples with inconclusive results found, whether they were included in the study and the AFIAS COVID-19 test result, if any

- line 140-142: this section reports viral load cut-offs that are not met in subsequent sections

Results

- Table 1b reported a classification of analyzed samples that does not meet the content of Methods section; moreover, there is no explanation of how and why Authors selected and calculated the cut-off and to which of target gene it refers (E or S or both target genes of molecular assay)

- Table 1a: an asterisk is reported near the values of prevalence, but there is no explanation of its meaning

- Table 1a: the number of negative samples is too high compared with PCR positives

- line 185-188: the population appears disproportionate between symptomatic and asymptomatic participants

- Table 2: Same considerations of Table 1b

- Table 3: Same considerations of Table 1b

6. PLOS authors have the option to publish the peer review history of their article (what does this mean?). If published, this will include your full peer review and any attached files.

Reviewer #1: No

Reviewer #2: No

Reviewer #3: **Yes: **Andrea Orsi

---

## [Author Response · Author response to Decision Letter 0]

20 Oct 2022

Point-by-point response to Reviewers and Academic Editor

Reviewer #1

Reviewer comment:

The reviewer in summarizing our work states:”… with the rapid antigen test AFIAS COVID-19 Ag, a fluorescence-based rapid antigen test. This is a not wide-spread rapid antigen test, not easy to work with as the development of the final reaction is fluorescence and a little instrumentation is required for reaction reading”

Response:

In reading what the reviewer writes, we realize that we have used a wrong wording in describing the characteristics of RADTs (i.e.“not requiring special equipment”). With "special equipment" (line 69 of the first manuscript version) we wanted to refer to a complex equipment to use, which required a particular laboratory training or particular technical skills. Furthermore, we also wanted to refer to the instrumentation footprint. Indeed, AFIAS COVID-19 Ag is very easy to work. The instrumentation for reading the test’s final reaction is of small dimension. Also, this test does not require a particular training of the laboratory personnel and the turnaround time is less than 20 minutes. For all these reasons, we state that AFIAS COVID-19 Ag is particularly suitable as a point of care test (https://www.youtube.com/watch?v=Up-ywSfpr9c ).

Reviewer comment:

The work is in line with many other papers on rapid antigen testing in COVID-19 pandemic showing that the sensitivity of these type of tests is higher in symptomatic than in asymptomatic patients and that using the Ct value as a proxy for viral load, the sensitivity increases with the increasing of viral loads. All these data have already been acknowledged by the scientific and medical community, the only reason to ask a re-submission from the authors is that there are not so many data on the type of rapid antigen test used by them, e.g. AFIAS COVID-19 Ag and it would be certainly useful to add more data to the existing ones.

Response:

We agree with the reviewer when she/he states that the results of our manuscript are in line with those of various other studies which have shown that the sensitivity of RADTs is generally higher in symptomatic than in asymptomatic subjects. However, we disagree with the reviewer when she/he says that the only reason for resubmitting our work would be the few data available on AFIAS. Unfortunately, the reviewer does not seem to take into account that in our study we also assessed the diagnostic accuracy of AFIAS based on the presence or absence of symptoms and on the level of exposure to SARS-COV-2.. The are very important aspects regarding the control of COVID-19, that is, the diagnosis and epidemiological surveillance (e.g. contact tracing) of the infection. These are very important aspects regarding the control of COVID-19, that is, the diagnosis and epidemiological surveillance (e.g. contact tracing) of the infection. Furthermore, the reviewer does not even take into account the particular type of study population that we considered in our study: a sample of the general population (i.e. mostly asymptomatic individuals) in a particular workplace, involving different infection exposures, subjected to SARS-COV-2 surveillance screening. 

Reviewer comment:

The good correlation with RT-PCR is only for Ct <22, which means that the technique of this rapid test e.g. fluorescence, is not as sensitive as the one of other rapid test working with fluorescence in the same context (Dinnes J, Deeks JJ, Adriano A, Berhane S, Davenport C, Dittrich S, et al. Rapid, point-of-care antigen and molecular-based tests for diagnosis of SARS-CoV-2 infection. Cochrane Database Syst Rev. 2020 Aug 26;8(8). There are no comments by the authors on this important issues.

Response:

The reviewer is right. Sensitivity and specificity data found for antigen immune-fluorescent assays considered in the systematic revision by Dinnes J (Cochrane Database Syst Rev, 2020) are higher than those found in our work for AFIAS. However, this is not at odds with the overall results of our study because the respiratory samples tested in the studies considered in that meta-analysis were from individuals with fever or with respiratory symptoms suggestive of COVID-19 (Weizel, 2020), or from suspected COVID-19 cases (Diao Clin Microbiol Infect 2021, 2021; Porte, Int J Infect Dis. 2020). Also in our study sensitivity was higher if analysis was restricted to samples from individuals with symptoms. It is important to note (as we have already stated om the discussion session of the first version of our manuscript) that in our study the larger proportion of NPSs was from asymptomatic participants, which accounted for 94.3% of the total NPSs and 63% of the RT-PCR positive. We have added a brief sentence in the discussion section to comment this issue.

Reviewer comment:

“Therefore, my suggestion is to re-write it as short report, making the study more synthetic, technical and readable, cutting down the number of tables to the essential (just one as the overall evaluation and one more according to the Ct level)”.

Response:

We prefer to maintain the current “long” format of the manuscript because re-writing it as a short report and especially cutting down the number of tables, as suggested by the reviewer (“just one as the overall evaluation and one more according to the Ct level”) would prevent us to correlate AFIAS accuracy with both presence or absence of COVID-19 symptoms and infection exposure, thus causing a loss of meaning and value in our work.

Reviewer comment:

The Ct level chosen for statistical analysis should be the same across all the study (e.g. abstract ct<25, table Ct< 22.34, text Ct <22), as there are only 49 positive samples by RT-PCR, 18 of them negative by the antigen test (the authors are working on 31 positive concordant samples, which is not that much). Therefore, there is no clue in making several subsets of positive samples/patients (and each one corresponds to a table!) since they are just few positive samples by the gold standard RT-PCR. This gives the work a strong reading difficulty and hard to follow the analysis.

Response:

We think that the fact that there are only 49 RT-PCR positive NPSs is not a sufficient and valid reason to adopt a single level of Ct for all statistical analysis in our work. This is because these 49 samples are not the same as regards clinical characteristics (presence of symptoms) and epidemiological characteristics (e.g. type and time of exposure to the virus) of the individuals from which they were collected. Studies investigating these aspects are important to support the use of RADTs in mass screening and epidemiological surveillance. In our opinion, the use of the median Ct for both PCR gene for each analysis is much more correct than the use of a unique cut-off for all analysis, and this approach has been used by other authors (Singh, Cereus 2021; Mak, J Clin Virol 2020; Ristic, PlosOne 2021; Singh, JMID 2021, Van der Moeren, Plos One2021). If we did not adopt the average median Ct levels specific for each type of sample subgroup, we would not be able to assess the influence of symptoms or exposure on the performance of theAFIAS. RT-PCR assays Ct values are relative, not absolute numbers. Ct values of the same RT-PCR assay varies between different labs. Furthermore, Ct values can vary up to two to three logs from test to test (Pritt, et al, 2020; https://www.publichealthontario.ca/-/media/documents/ncov/main/2020/09/cycle-threshold-values-sars-cov2-pcr.pdf?la=en;
https://perkinelmer-appliedgenomics.com/2021/03/04/covid-19-rt-pcr-ct-values/

Reviewer comment:

Moreover, it is not clear if the statistical analysis is run on sample or patients (1294 vs 1183, text vs. tables).

Response

 We analyzed 1294 samples belonging to 1183 participants (line 158 and line 162 of the first manuscript version). To avoid further misunderstandings, in revising the manuscript, we used the term “nasopharyngeal swabs (NPSs)” instead of the generic term "samples".

Reviewer #2

The authors evaluated a SARS-CoV-2 rapid antigen detection test (RADT) while using RT-PCR as the gold standard. The works were scientifically sound. However, mistakes and confusions were found and I have several queries for the current version of the manuscript. I hope the authors will find them useful to revise it so that the quality could be improved.

Reviewer comment:

- the research gap was not well defined:

(1) lines 95-100: the authors have to explain the reasons for choosing ‘military and civilian personnel of a military airport in the metropolitan area of Rome’ as the subjects

(2) lines 69, 87-88, 114-121: the authors stated that RADTs are easy to use since equipment was not required, however, the authors employed the fluorescence based RADT in the study. This test requires an equipment to read the test results. It seems that the introduction was discordant to the method used.

Response:

(1) The study was part of the public health response to control as soon as possible any outbreak occurring in the military airport (as reported in the Scientific Collaboration Protocol signed by the Experimental Flight Center, Italian Air Force Logistic Command and Istituto Superiore di Sanità on 30 November 2020), with simultaneous evaluation of the RADT used for the screening.

(2) In reading this comment, we realize that we have used a wrong wording in describing the characteristics of RADTs (i.e.“not requiring special equipment”). With "special equipment" (line 69 of the first manuscript version) we wanted to refer to a complex equipment to use, which required a particular laboratory training or particular technical skills. Indeed, AFIAS COVID-19 Ag is very easy to work with. The instrumentation for reading the test’s final reaction is of small dimension. Also, this test does not require a particular training of the laboratory personnel and the turnaround time is less than 20 minutes. For all these reasons we state that AFIAS COVID-19 Ag is particularly suitable as a point of care test (https://www.youtube.com/watch?v=Up-ywSfpr9c).

Reviewer comment:

line 172: you have to mention the meaning of the cut-off value Ct 22.34, readers will not know

that unless they went through the table 1b

Response:

The reviewer is right. In the revised version of the manuscript we have clarified the meaning of the cut-off value Ct 22.34.

Reviewer comment:

lines 220-221, 272-273, table 3, footnote a: you have to define group B and group C clearly. I still do not know the differences between groups B and C after going through these explanations. I only know that participants in groups B and C had contact history with COVID-19 patients within the last 14 days. Participants in groups B will be more likely to be detected by RADT than those in group C. If I understand correctly, groups B participants had contact history with COVID-19 patients tested positive by RADT while groups C participants had contact history with COVID-19 patients tested positive by RT-PCR. Please confirm my speculation.

Response:

 We apologize to the reviewer. We actually failed to define groups B and C clearly enough. Let's try to do that now: group B participants had contact history with COVID-19 case confirmed by RT-PCR; group C participants had contact history with individual tested positive for SARS-CoV-2 by a molecular or an antigenic test (by answering the question in the questionnaire, participants were unable to specify whether the test was antigenic or molecular). In the revised version of the manuscript we defined more clearly the groups B and C. 

Reviewer comment:

the authors analyzed the RADT results according to the four different parameters: (1) EU HSC (2) RT-PCR results (3) presence of symptoms (4) contact history with COVID-19 cases. It means that many different cut-offs were used:

(1) EU HSC cut-offs: <25, 26-30, 30-36, >36 (lines 141-142)

(2) median RT-PCR: 22.34

(3) symptomatic patients: 18.08

(4) asymptomatic patients: 26.00

For (1), a table is preferred rather than just describing the results in text only (lines 179-182).

Response:

We think that the use of the median Ct for both PCR is the most correct procedure to analyse AFIAS accuracy in each subgroup of our study population. This is because the 49 RT- PCR positive participants in our study are different as regards clinical characteristics (presence of symptoms) and epidemiological characteristics (type and time of exposure to the virus). If we did not adopt the average median Ct levels specific for each type of population subgroup, we would not be able to assess the influence of symptoms or exposure on the performance of the test. The median Ct for both PCR gene as cut-off for accuracy analysis has also been adopted by other authors. (e.g. Singh, Cereus 2021; Mak, J Clin Virol 2020; Ristic, PlosOne 2021; Singh, JMID 2021, Van der Moeren, Plos One2021). However, RT-PCR assays Ct values are relative, not absolute numbers. Ct values of the same RT-PCR assay varies between different labs. Furthermore, Ct values can vary up to two to three logs from test to test (Pritt, et al, 2020; https://www.publichealthontario.ca/-/media/documents/ncov/main/2020/09/cycle-threshold-values-sars-cov2-pcr.pdf?la=en;
https://perkinelmer-appliedgenomics.com/2021/03/04/covid-19-rt-pcr-ct-values/)

Anyway, we added a new Table (Table 2) for the EU HSC cut-offs. 

Reviewer comment:

Minor comments:

- avoid creating unnecessary abbreviations if the fluency is not improved, it is not inconvenient to spell the terms SS (line 73) and SP (line 75) in full as sensitivity and specificity respectively. I cannot see those terms will either save the word counts or make the manuscript more neat and tidy. In addition, these two terms are not commonly used by other research groups. Readers have to memorize them throughout the manuscript. It is easy to create confusion. On the other hand, the term ‘N’ referring nucleocapsid protein created (line 117) has not been used in subsequent texts. All these kinds of arrangements make this manuscript quite unprofessional.

- there were only two different assays, RT-PCR and RADT, it is not necessary to create another term, index test in lines 34, 41, 44, 49, 169, 200, 211, 254, 260. As you define RADT at the beginning, you can either use this term throughout the manuscript or make a short form for the RADT that your performed in your study, ‘AFIAS’ to refer ‘AFIAS COVID-19 Ag’.

Response:

The reviewer is right. In the revised manuscript version, we have deleted the abbreviation for nucleocapsid protein (N) and we have spelled in full sensitivity and specificity. Also, after the first citation we have used AFIAS’ to refer ‘AFIAS COVID-19 Ag throughout the manuscript.

Reviewer #3

Reviewer comment:

The study was conducted appropriately. Nevertheless, an important limitation is that the study was carried out almost a year and a half ago: the epidemiological situation and the variants circulating today are different. 

Response:

The impact of variability of nucleocapsid region in circulating variants on performance of RADT test is carefully monitored in the world. To date, in our knowledge, no evidence has been reported about changes significantly influencing the sensitivity of RADTs.

Reviewer comment:

Moreover, the topic of the article appears to be of limited interest since it has been extensively covered in similar published works regarding the same rapid antigen test and others similar. In addition to the two studies already mentioned in the manuscript’s discussion, some other examples are reported below:

- Baccani I, Morecchiato F, Chilleri C, Cervini C, Gori E, Matarrese D, Bassetti A, Bonizzoli M, Mencarini J, Antonelli A, Rossolini GM. Evaluation of Three Immunoassays for the Rapid Detection of SARS-CoV-2 antigens. Diagn Microbiol Infect Dis. 2021 Oct;101(2):115434. doi: 10.1016/j.diagmicrobio.2021.115434. Epub 2021 May 21. PMID: 34174523; PMCID: PMC8137375.

- Parvu, V.; Gary, D.S.; Mann, J.; Lin, Y.C.; Mills, D.; Cooper, L.; Andrews, J.C.; Manabe, Y.C.; Pekosz, A.; Cooper, C.K. Factors that influence the reported sensitivity of rapid antigen testing for SARS-CoV-2. Front. Microbiol. 2021, 12, 714242.

- Filchakova, O.; Dossym, D.; Ilyas, A.; Kuanysheva, T.; Abdizhamil, A.; Bukasov, R. Review of COVID-19 testing and diagnostic methods. Talanta 2022, 244, 123409.

- Bruzzone, B.; De Pace, V.; Caligiuri, P.; Ricucci, V.; Guarona, G.; Pennati, B.M.; Boccotti, S.; Orsi, A.; Domnich, A.; Da Rin,G.; et al. Comparative diagnostic performance of rapid antigen detection tests for COVID-19 in a hospital setting. Int. J. Infect.Dis. 2021, 107, 215-218.

Response:

We are aware that the topic of our study has been covered in many other similar studies. In the study by Baccani et al. who also used AFIAS COVID-19 Ag, only 81 samples, collected from known positive and negative COVID-19 patients, were tested. The very small study sample was the reason because we did not cite that work. The other studies cited by the reviewer were also conducted on known positive and negative SARS-CoV-2. We would like to stress that our study is one of the not numerous (at all) accuracy studies in which the study population consists of general people, therefore largely asymptomatic. We think studies like ours can be useful to support the use of these tests in mass screening and epidemiological surveillance. 

In the revised version of the manuscript, we have added the studies cited by the reviewer

Reviewer comments:

Here below, Authors can find a list of revisions that need to be addressed in order to improve the quality of the manuscript.

Abstract

- line 38: median age is missing from results

Response:

Median age was 42 years. We have added this data.

- line 41-44: it is not so clear what concept the authors want to express

Response:

The reviewer is right. The sentence is misspelled. In the revised version of the manuscript we rewrote that sentence as follows: “AFIAS sensitivity tended to be higher for NPSs with higher viral load. A higher sensitivity (0.944) compared to the overall baseline sensitivity (0.633) was also found for NPSs from participants with COVID-19 compatible symptoms, for which K was 0.891 (almost perfect).

Reviewer comment:

Table 1b reported a classification of analyzed samples that does not meet the content of

 Methods section;

Response: The reviewer is right. In the Methods section of the revised version of the manuscript we have added some sentences which clarify what is reported in table 1b. 

Reviewer comment:

Methods

- the method of enlistment is not specified (they were volunteers?)

Response: They were not enlisted on a voluntary basis. The study was part of the public health response to control as soon as possible any outbreak occurring in the military airport with simultaneous evaluation of the RADT used for the screening. This information is included in Methods and Ethical Approval Statement 

- It is not specified how was determined the number of people to be enlisted (only time criteria?)

Response: The study population included people screened from November 2020 to April 2021, to control the transmission in the workplace.

Reviewer comment: 

- line 104-106: we suggested to specify the use that will be made of the data contained in the questionnaires

Response:

We have specified the use of the data collected through the questionnaire.

Reviewer comment:

- viral variants circulating at the time of the study are never mentioned in the study. Therefore, data on their detection capabilities for the test under consideration are missing.

Response: The impact of variability of nucleocapsid region in circulating variants on performance of RADT test is carefully monitored in the world. To date, in our knowledge, no evidence has been reported about changes significantly influencing the sensitivity of RADTs.

Reviewer comment:

- line 139-140: it would be helpful to specify the number of samples with inconclusive results found, whether they were included in the study and the AFIAS COVID-19 test result, if any

Response: Actually, no sample with inconclusive result (no detectable signal for both target genes and internal control) was observed. 

Reviewer comment:

- line 140-142: this section reports viral load cut-offs that are not met in subsequent sections

Response

In the revised version of the manuscript these viral load Ct cut-offs were included in the new table 2.

Reviewer comment: 

Results

- Table 1b reported a classification of analyzed samples that does not meet the content of Methods section; moreover, there is no explanation of how and why Authors selected and calculated the cut-off and to which of target gene it refers (E or S or both target genes of molecular assay)

Response:

The reviewer is right. In the Methods section of the revised version of the manuscript, information on classification of the different sample groups (study population’s paragraph) and on the cut-of calculation (SARS-CoV-2 Molecular Detection paragraph) have been given.

Reviewer comment:

Table 1a: an asterisk is reported near the values of prevalence, but there is no explanation of its meaning

Response:

The reviewer is right. It was a typo that escaped later checks.

Reviewer comment:

- Table 1a: the number of negative samples is too high compared with PCR positives.

Response:

We think this is normal in a population like the one that was tested in our study. The prevalence of positive RT-PCR samples was 3.8%, which is perfectly in line with that recorded throughout Italy in that period. We must reiterate that our study population consisted of the general population and about 94% of the participants were asymptomatic.

Reviewer comment:

- Table 2: Same considerations of Table 1b

- Table 3: Same considerations of Table 1b

Response:

The reviewer is right. In the Methods section of the revised version of the manuscript we have added some sentences which clarify what is reported in table 1b.

Response to Academic Editor

Editor comment:

The work described in this manuscript is valid even though the study is not original and the results were predictable based on previous literature. However, the manuscript should be revised for clarity, internal congruency and completeness of details and information. The reviewers identified critical issues and concerns that should be addressed point-by-point in the revised manuscript. If you decide to re-submit the manuscript, I'd also invite you to consider reducing the number of tables as suggested by one reviewer. 

Response

We are aware that the topic of our study has been covered in many other similar studies. However, we would like to stress that our study is one of the not numerous (at all) accuracy studies in which the study population consists of general people, therefore largely asymptomatic. Besides, our study was carried out in a particular workplace, involving different infection exposures, subjected to SARS-COV-2 surveillance screening. We think studies like ours can be useful to support the use of these tests in mass screening and epidemiological surveillance. 

As regards the response to the reviewers, we think that all of the requests and suggestions of both Reviewer #2 and Reviewer #3 have been met. Regarding Reviewer #1, while trying to answer point by point to his comments, we regret to say that we disagree with either his overall judgment of the study or the suggestion to rewrite it as a short report and cutting down the number of tables to the essential (“just one as the overall evaluation and one more according to the Ct level”). Following this reviewer's suggestion would prevent us to assess AFIAS accuracy according both presence or absence of COVID-19 symptoms and infection exposure, thus causing a loss of meaning and value in our work. We think the analysis of these aspects are crucial to support the use of RADTs in mass screening and epidemiological surveillance.

We replaced the previous information in Funding source and Financial disclosure with the following statement: “The study was supported by the institutional fund of the Italian National Institute of Health (Istituto Superiore di Sanità) in the frame of its institutional role in public health activities for the COVID-19 emergency (internal code for administrative management: Fascicolo BA17).”

Resubmitting the manuscript, we uploaded our minimal underlying data set as Supporting Information files.

---

## [Decision Letter · Decision Letter 1]

31 Oct 2022

PONE-D-22-13832R1Diagnostic accuracy of a SARS-CoV-2 rapid antigen test among military and civilian personnel of an Air Force airport in central ItalyPLOS ONE

Dear Dr. Spada,

Thank you for submitting your manuscript to PLOS ONE. After careful consideration, we feel that it has merit but does not fully meet PLOS ONE’s publication criteria as it currently stands. Therefore, we invite you to submit a revised version of the manuscript that addresses the points raised during the review process. Please re-consider the description of fluorescence-based RADT. 

We look forward to receiving your revised manuscript.

Kind regards,

Etsuro Ito

Academic Editor

PLOS ONE

Journal Requirements:

Reviewers' comments:

Reviewer's Responses to Questions

**Comments to the Author**

1. If the authors have adequately addressed your comments raised in a previous round of review and you feel that this manuscript is now acceptable for publication, you may indicate that here to bypass the “Comments to the Author” section, enter your conflict of interest statement in the “Confidential to Editor” section, and submit your "Accept" recommendation.

Reviewer #2: (No Response)

2. Is the manuscript technically sound, and do the data support the conclusions?

Reviewer #2: Yes

3. Has the statistical analysis been performed appropriately and rigorously? 

Reviewer #2: Yes

4. Have the authors made all data underlying the findings in their manuscript fully available?

Reviewer #2: Yes

5. Is the manuscript presented in an intelligible fashion and written in standard English?

Reviewer #2: Yes

6. Review Comments to the Author

Reviewer #2: The authors addressed all of my queries, the revised version is better than the previous one. I went through the reviewer 1 and reviewer 3 comments and I shared similar views for some of them. I have the following three comments, the quality of the manuscript can be improved further.

1. The authors should elaborate more about the background of the study by copy and paste your response ‘The study was part of the public health response to control as soon as possible any outbreak occurring in the military airport (as reported in the Scientific Collaboration Protocol signed by the Experimental Flight Center, Italian Air Force Logistic Command and Istituto Superiore di Sanità on 30 November 2020), with simultaneous evaluation of the RADT used for the screening.’ into line 95, after the sentence ‘………….transmission of SARS-CoV-2 infection in the workplace.’

2. Reviewer 1 and I shared similar concerns of using fluorescence based RADT in your study, it is worthwhile to justify the reasons for selecting this RADT instead of traditional RADT without using fluorescence instrument.

3. Both reviewers 1 and 3 raised the concerns of similar studies have been published between 2020 and 2022. I also shared this view when I reviewed the first version, however, the PLOS journal did not focus on the novelty of the research. As I am not the new reviewer in PLOS, that’s why I did not raise this out. I suggest you should go through your manuscript thoroughly, try to write more about the background and the reasons to launch this study. The above comments 1 and 2 are aimed at filling this gap.

7. PLOS authors have the option to publish the peer review history of their article (what does this mean?). If published, this will include your full peer review and any attached files.

Reviewer #2: No

---

## [Author Response · Author response to Decision Letter 1]

3 Nov 2022

Point-by-point response to Reviewers

We thank Reviewer #2 for her/his further requests: they allowed us to improve further the manuscript.

Reviewer #2

Reviewer comment 1:

1. The authors should elaborate more about the background of the study by copy and paste your response ‘The study was part of the public health response to control as soon as possible any outbreak occurring in the military airport (as reported in the Scientific Collaboration Protocol signed by the Experimental Flight Center, Italian Air Force Logistic Command and Istituto Superiore di Sanità on 30 November 2020), with simultaneous evaluation of the RADT used for the screening.’ into line 95, after the sentence ‘………….transmission of SARS-CoV-2 infection in the workplace.’

Response:

The sentence was copy-pasted in the manuscript (line 95, Methods - Study population section) as suggested by the reviewer (line 100 in the revised version)

Reviewer comment 2:

2. Reviewer 1 and I shared similar concerns of using fluorescence based RADT in your study, it is worthwhile to justify the reasons for selecting this RADT instead of traditional RADT without using fluorescence instrument.

Response:

Although the test requires a fluorescence reader, the running time from sample to result is comparable to traditional RADTs; however, it provides the added value that the result is not subject to the operator's interpretation. In fact, the positive/negative output of the fluorescence reader is automatic, based on an algorithm comparing the fluorescence obtained from the sample to a cut-off value; in the traditional RADTs, the operator can see more or less well a faint colored band of positivity

We included this explanation in the “Introduction” section of the manuscript (line 84, after “… in human NPSs within 20 minutes.”)

Reviewer comment 3:

3. Both reviewers 1 and 3 raised the concerns of similar studies have been published between 2020 and 2022. I also shared this view when I reviewed the first version, however, the PLOS journal did not focus on the novelty of the research. As I am not the new reviewer in PLOS, that’s why I did not raise this out. I suggest you should go through your manuscript thoroughly, try to write more about the background and the reasons to launch this study. The above comments 1 and 2 are aimed at filling this gap.

Response:

The manuscript was revised and modified in the “Introduction” and “Methods” section, according to the reviewer comments 1 and 2; in particular, we stressed that:

- the use of a fluorescence reader makes results less dependent from the operator interpretation than traditional tests

- the study population is unselected and mostly asymptomatic

- the study was carried out in the frame of a public health response, with simultaneous evaluation of the RADT used for screening

---

## [Editor Report · Decision Letter 2]

6 Nov 2022

Diagnostic accuracy of a SARS-CoV-2 rapid antigen test among military and civilian personnel of an Air Force airport in central Italy

PONE-D-22-13832R2

Dear Dr. Spada,

We’re pleased to inform you that your manuscript has been judged scientifically suitable for publication and will be formally accepted for publication once it meets all outstanding technical requirements.

Kind regards,

Etsuro Ito

Academic Editor

PLOS ONE

---

## [Editor Report · Acceptance letter]

15 Nov 2022

PONE-D-22-13832R2 

Diagnostic accuracy of a SARS-CoV-2 rapid antigen test among military and civilian personnel of an Air Force airport in central Italy 

Dear Dr. Spada:

I'm pleased to inform you that your manuscript has been deemed suitable for publication in PLOS ONE. Congratulations! Your manuscript is now with our production department. 

Kind regards, 

on behalf of

Prof. Etsuro Ito 

Academic Editor

PLOS ONE